# CAN - CONTINUOUSLY ADAPTING NETWORKS

## ABSTRACT

Catastrophic forgetting is a fundamental challenge in neural networks that prevents continuous learning, which is one of the properties essential for achieving true general artificial intelligence. When trained sequentially on multiple tasks, conventional neural networks overwrite previously learned knowledge, hindering their ability to retain and apply past experiences. However, humans and other animals can learn new things continuously without forgetting them. To overcome this problem, we devised an architecture that preserves significant task-specific connections by combining selective neuron freezing with Hebbian learning principles. Hebbian learning enables the network to adaptively strengthen synaptic connections depending on parameter activation. It is inspired by the synaptic plasticity seen in the brain. By preserving the most important neurons using selective neuron freezing, new tasks can be trained without changing parameter weights. Experiments conducted on standard datasets show that our model significantly reduces the risk of catastrophic forgetting, allowing the network to learn continually.

## 1 INTRODUCTION

Artificial Neural Networks (ANNs) are an important aspect of modern Deep Learning models. Even though they are biologically plausible in certain aspects, they are still incapable of doing many important tasks that biological neural networks can easily perform. Continual Learning is one area where ANNs don't work well with their vanilla architecture Wang et al. (2024). To learn tasks from a new distribution, ANNs have to be trained on the entire dataset from the beginning due to which they end up forgetting the previously learned task Zenke et al. (2017). On the other hand, the human brain can learn from non-stationary data and store it easily Mao et al. (2021); Kudithipudi et al. (2022). It could be because of poor mimicry of the biological capabilities of the neurons that ANNs currently don't have this property, or it could also be because of some missing piece in the architecture Bashivan et al. (2019).

Catastrophic forgetting is a problem faced by the current designs of ANNs where the network forgets the previously learned data distribution when trained on different data distributions Bagus & Gepperth (2021); De Lange et al. (2022); Toneva* et al. (2018); Harun et al. (2023). Continual Learning is an important and interesting property for a neural network to attain. There are a few different methods using which other papers have tried to solve this problem Zenke et al. (2017); Yang et al. (2023). We have tried to solve this issue through architectural changes in this work. In general ANNs, the parameters are trained to store the tuned values corresponding to the tasks. Updating these parameters for a different distribution makes the model forget the previous task. Therefore, the most intuitive way to retain the memory of the previous task is to freeze those parameters Hou et al. (2025). Then comes the next intuitive question, i.e., which parameters to freeze? We developed a method that utilizes Hebb's principle to compute a hebbian update matrix which upon normalization acts as scaler values to scale the incoming gradients with the connection importance. This allows the algorithm to strengthen or weaken or completely ignore the gradients based on the connection importance. As new tasks keep coming, it is made sure that the previously trained neurons are not updated. The key highlight of this algorithm is that it dynamically selects the important neurons relevant for the task based on the incoming data distribution and trains accordingly Amato et al. (2019). The idea inspired from how the brain uses both local and global feedback to optimize and update the neurons connections.

In this paper, we will show how a general ANN can be extended to attain Continual Learning by dynamically training sub-networks for different tasks Kohonen (1982). We show this is possible using a simple combination of local and global feedback systems.

## 2    RELATED WORK

Continual Learning has been addressed through multiple approaches in the past. Here are some of the techniques using which this was done:

### 2.1    REGULARIZATION-BASED METHODS

Regularization techniques use penalties to limit changes to important parameters during the learning of new tasks. These approaches retain task-specific knowledge by minimizing parameter updates that could damage previously learned knowledge. One notable method is Synaptic Intelligence (SI), which stores task-relevant information over time to prevent catastrophic forgetting Zenke et al. (2017). Other approaches use elastic weight consolidation (EWC) or L2 regularization to maintain network stability while learning new tasks Kann et al. (2023).

### 2.2    REPLAY-BASED METHODS

Replay-based approaches store samples from previous tasks in memory and periodically revisit them during training on new tasks to maintain learned knowledge van de Ven et al. (2020). The stored samples can be used for rehearsal, ensuring the model retains information from earlier tasks. Recent works have shown that even simple memory replay methods can outperform more complex strategies when memory is managed effectively Bagus & Gepperth (2021). Techniques like generative replay have also been explored Bashivan et al. (2019), where synthetic samples from previous tasks are generated and used for rehearsal. Certain techniques make use of external memory to store and fetch required data Du et al. (2022); Sun et al. (2024).

### 2.3    PARAMETER ISOLATION TECHNIQUES

Parameter isolation methods assign different sets of parameters to different tasks, preventing interference between them. A prominent example is the Progressive Neural Network, which grows the network by adding new neurons for each task, avoiding overwriting the learned parameters of previous tasks. Freezing critical parameters, based on neuron importance scores, is another approach to prevent interference Hou et al. (2025). This technique can be effective but may lead to inefficiencies as networks expand.

### 2.4    DYNAMIC ARCHITECTURES

Dynamic architectures allow the model to grow and adapt based on the complexity of the tasks. This approach modifies the network's architecture by adding or adjusting components as new tasks are introduced, making it more flexible in dealing with non-stationary data. Recent work using Hebbian Learning has proposed dynamic architectures that automatically freeze important neurons and connections, allowing the model to learn incrementally without catastrophic forgetting Amato et al. (2019). Another recent work using probabilistic modeling has proposed approaches like Continual Learning with Adaptive Weights (CLAW), which adaptively determine shared and task-specific components to mitigate catastrophic forgetting while enabling task transfer Adel et al. (2020).

### 2.5    META-LEARNING APPROACHES

Meta-learning, or learning to learn, has been applied to create models that quickly adapt to new tasks with minimal forgetting. These methods often involve training the model on task sequences, enabling it to generalize well to unseen tasks while retaining knowledge from prior ones. Meta-learning-based Continual Learning has shown promise in effectively balancing stability and plasticity Zenke et al. (2017); Javed & White (2019).

## 3  METHODOLGY

In order to solve the Continual Learning problem, we need to stop the relevant parameters of the previous tasks from changing while training for a newer task. This can be achieved by calculating an "importance score" for each parameter. One of the ways to calculate this importance score is to use a locally learning algorithms that computes a score using the activations of all the parameters of a neuron. One such algorithm is Hebbian learning. Since this algorithm is really good at learning patterns and it dynamically learns patterns according to the coming tasks, it can be applied to any dataset, making it adaptable for dynamically identifying important neurons across different tasks. However, it is crucial to ensure that previously trained neurons are not selected again during the training of new tasks. We have designed an architecture where the gradients of the SGD algorithm is scaled by the locally received feedback. This feedback is computed using the Hebbian learning algorithm.

This model starts by training a set of parameters $\boldsymbol{\theta}_1$ for a given task and subsequently training another set of parameters $\boldsymbol{\theta}_2$ for the next task. Here, both $\boldsymbol{\theta}_1$ and $\boldsymbol{\theta}_2$ belong to the set of parameters $\boldsymbol{\theta}$. The goal of the network is to minimize the loss for a task $\boldsymbol{T}_n$ using the parameters $\boldsymbol{\theta}_n$ which belongs to the set $\boldsymbol{\theta}$.

$$L(\boldsymbol{\theta}) = L_n(\boldsymbol{\theta}_n) \in \boldsymbol{T}_n$$

The model learns to select important parameters for a given task and train them for that particular task. The important neurons are selected dynamically using the computed importance score. These scores are calculated using Hebbian Learning Bahroun & Soltoggio (2018). The learning rule for Hebb's Principle is: "Neurons that fire together wire together". This is considered the property of the human brain and is responsible for the plasticity of the connections. Frequently activating the relevant connections allows us to learn the action for that particular task.

### 3.1  HEBBIAN LEARNING

#### 3.1.1  INTRODUCTION

Hebbian Learning is an unsupervised algorithm that updates the parameters according to their activation. Since activations are different for different kinds of data, the important parameters to be updated change according to activated values. In order to calculate an importance score for every neuron, we calculate the average of all the weights relevant to one particular neuron at a time and repeat the same for all the neurons. At the end, the values are scaled to get a score value. The Hebbian weights are updated using the following formula :

$$\delta w = lr * (y_i * x_j)$$

Where $w$ is the Hebbian parameter, $lr$ is the learning rate, and $x$ and $y$ are input and output values from the synapses, respectively. Hebbian Learning works locally by depending only on the input and output values of the synapses. This means that it doesn't rely on any global error signal and works for unseen data. It naturally strengthens connections between co-active neurons, allowing the network to find relevant connections for a particular data distribution.

#### 3.1.2  OJA'S RULE

The base form of Hebbian learning doesn't really normalize the weights but rather continuously keeps on increasing the parameter if the connection is strong every time. In other words, the parameters very easily explodes and reaches infinity. Oja's Rule stabilizes hebbian learning with a very simple way. To update rule is as follows:

$$\delta w = lr * y_i * (x_j - y_i * w_{ij})$$

Oja's rule normalizes the weights and can be used in practice to find the first principal component, effectively performing PCA. The resulting weight vector coincides with the principal axis. A single linear neuron trained with Oja's rule thus extracts the most information it can from the input samples.

### 3.1.3 LATERAL INHIBITION

The aforementioned learning rules of Hebbian learning are only applicable to networks with a single output neuron. Every single neurons learns the first principal component, but this is not what we want. It is expected for each of the neurons to learn different patterns to give prediction on all inputs. Therefore we introduce competition among the neurons. Each neuron can now inhibit every other neuron (in the same layer) and competition is established. When a certain input pattern is applied, the neurons compete and only a few neurons respond strongly to the pattern, while the other neurons are inhibited. Such an inhibition scheme is actually biologically motivated. This encourages different neurons to learn various patterns.

## 3.2 SELECTIVE TRAINING AND INFERENCE MASKING

Selective training is used to train only the selected neurons by freezing the others. This prevents the loss of previously trained tasks. In PyTorch, we implemented hooks that automatically multiply the scaled hebbian updated values with the gradients. In such a method, gradients are scaled according to the importance of the connections and even completely ignored if the scaling value is 0. During the forward propagation, we use a masking technique where the activation from irrelevant neurons for a particular task is multiplied by zero. This allows the data to flow only through the few selected neurons and train them.

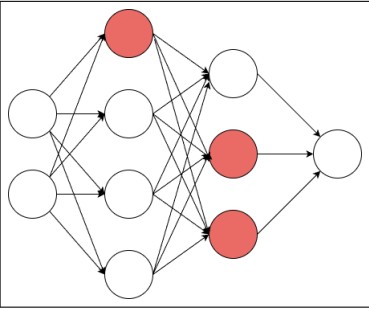

*Figure 1: The red neurons represent the rejected neurons. While training, the gradient only flows through the white neurons and only trains them. The red neurons only receive zero gradients until they become relevant for the next task. The gradients of the other relevant neurons are scaled according to it's importance.*

Figure 1 shows how important neurons are selected and others are masked. The red ones are the rejected neurons and the remaining neurons get together to form a sub-network. These sub-networks are dynamically formed even with the use of non-stationary data.

Every task will have its sub-network and is stored as masks in the db. During the inference time, a relevant mask can be selected and used to get the prediction. Now, the question arises of how we choose an appropriate mask. We are also working on creating a gating mechanism, which is part of this project's future scope.

## 3.3 ARCHITECTURE

Our architecture comprises three important modules: the ANN, Importance Calculator and Selective Trainer. Importance Calculator uses Hebbian Learning, an unsupervised algorithm to dynamically and locally calculate scores. Based on these scores, the Selective Trainer trains only those neurons with a score above a pre-defined threshold. The Trainer also trains the selected neurons with the received local feedback from the hebbian update. It simply scales the gradients of SGD with the connection importance.

Importance Calculator uses Hebbian Learning, an unsupervised algorithm to dynamically and locally calculate scores. Hebbian Learning is used here since it can learn important representation from non-stationary data, especially unsupervised. These scores are then passed to the Selective Trainer, a function that uses hooks to select relevant neurons based on their scores and scale the

remains ones. It does this by setting the gradient of irrelevant neurons to zero during the training process, thereby focusing the model's Learning on the most important neural pathways.

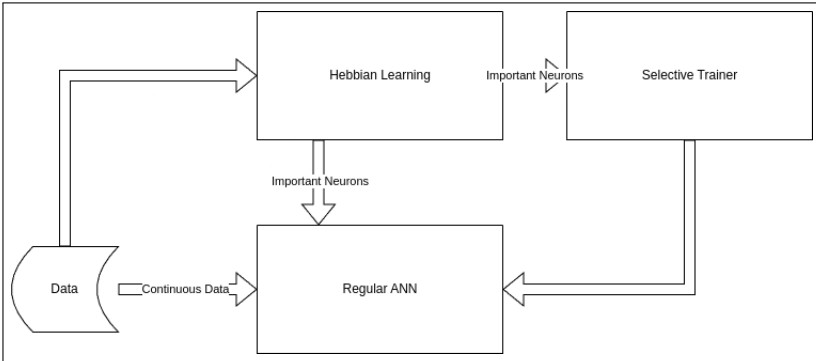

*Figure 2: Data Flow Diagram of CAN's Architecture.*

Figure 2 shows the Data Flow diagram of our architecture. This shows how different important component of our architecture work together to solve the problem of Continual Learning.

## 4 EXPERIMENT

All our experiments were conducted on the MNIST and the CIFAR-10 Dataset. We tested our model for both task-incremental learning and domain-incremental learning. All our experiments were run on a network with hidden layers as 256, 128 and 64 with ReLU activation functions and Sigmoid activation at the end.

### 4.1 DATASET

#### 4.1.1 TASK-INCREMENTAL LEARNING

For our experiments, we divided the MNIST and the CIFAR-10 Dataset into two sets with the classes 0-4 and 5-9 respectively. Experiments could be conducted with any such classification data by dividing them into multiple tasks. One of the constraints of our architecture is that we can't use a continuous stream of data belonging to a variable number of classes.

The division of data doesn't need to be equal all the time. Task-Incremental learning measures the amount of forgetting the network goes through after training the model on different tasks.

#### 4.1.2 DOMAIN-INCREMENTAL LEARNING

For this experiment, we create two different version of the same dataset. The first version is the normal training data as it is and second version is added with different random properties like HorizontalFlip, RandomRotation, RandomAffine, ColorJitter, random noise.

The model is then trained on both the datasets one by one. In this case masks are not used for forward propagation as the entire dataset is used when it is trained both the times. Our algorithm showed improvements from the vanilla model.

### 4.2 TRAINING

#### 4.2.1 TASK-INCREMENTAL LEARNING

The model is first trained on classes 0-4. The model automatically identifies the most activated neurons and trains the regular gradients scaled with the importance factor of the connection. The model is trained for 10 epochs with a learning rate scheduler to attain stability. After the first training, the mask representing the sub-network for this particular task is stored as a binary mask. This mask

is then used for both further training and inference. While training for the subsequent tasks, the previous masks are used to ensure that those neurons are not affected during further training.

The model is then trained on classes 5-9, using the same hyper-parameters as in the previous training phase. Simultaneously, it ensures that no neurons selected for the current training were used for prior tasks. This training approach can be generalized with $N$ number of tasks. The network will continue learning tasks until it reaches its capacity, meaning all neurons have been utilized. In such a situation, the concept of a growing network comes into the picture. Implementing this is part of the project's future work.

### 4.2.2 DOMAIN-INCREMENTAL LEARNING

The model is trained on the regular dataset first. The model uses our algorithm which uses the combination of gradient descent and hebbian learning. In this experiment there are no masks since the entire data is used to train. The training with the second dataset then starts with a similar fashion the first one.

After the training, the model is tested on the test data of both the datasets one by one to compute the average accuracy.

### 4.3 INFERENCE

For the task-incremental experiment, After the training phase, there exists a binary mask for each of the task. Currently, to analyze the performance of the model, we are manually selecting the mask and measuring the metrics but it can be done using a gating system that automatically selects the relevant gate according to the given task during inference. A gating system can be easily implemented by training a simple auto-encoder for each task. During the inference phase, data can be passed through every auto-encoder to calculate the reconstruction error and hence choose the auto-encoder with the least error. The mask corresponding to that particular auto-encoder would be used for the inference.

Whereas for the domain-incremental experiment, we just simply test the model on the test sets of both the datasets one by one.

## 5 RESULTS

We have measured the capabilities of our model using the average accuracy metric. We have plotted some graphs to make the differences clear between our model and the vanilla one.

### 5.1 TASK-INCREMENTAL LEARNING & DOMAIN-INCREMENTAL LEARNING RESULTS

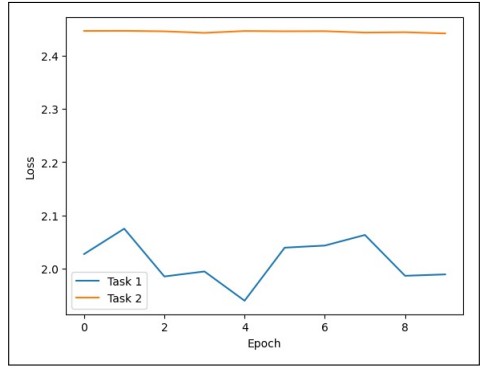

*Figure 3: The Vanilla network running regular SGD performs as follows in the Task-Incremental Task*

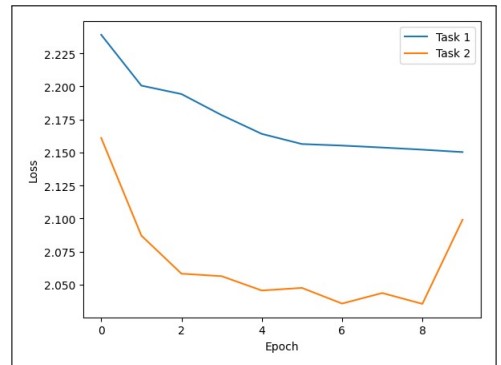

*Figure 4: The network running our algorithm performs as follows in the Task-Incremental Task*

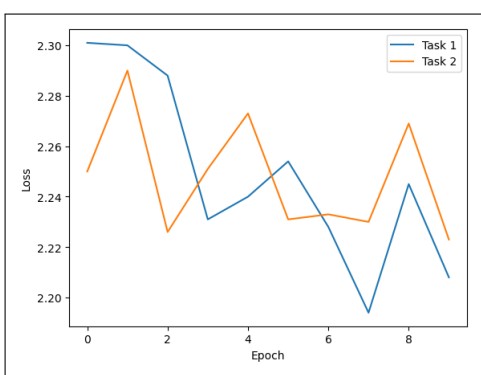

*Figure 5: The Vanilla network's performance in the domain-incremental task.*

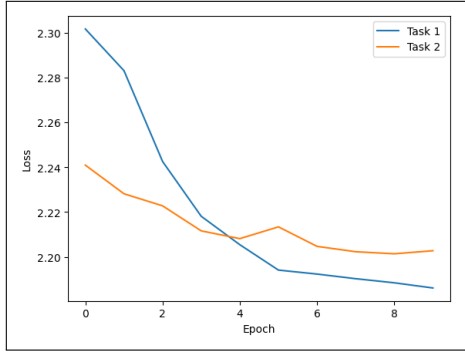

*Figure 6: The network's performance in the domain-incremental task using our algorithm*

## 5.2 AVERAGE ACCURACY (ACC)

It is the average accuracy over all tasks after the entire sequence of tasks has been learned. It provides a summary of how well the model performs across the different tasks once the final task has been completed. This metric offers a holistic view of the model's effectiveness in Continual Learning. A high ACC indicates that the model successfully learns new tasks while retaining knowledge of older tasks.

$$\text{ACC} = \frac{1}{n} \sum_{i=1}^{n} A_{i,n}$$

Where $A_{i,n}$ is the accuracy on task $i$ after learning the last task $n$ and $n$ is the total number of tasks.

*Table 1: Average Accuracy*

| MODEL | AVG ACC of TIL | AVG ACC of DIL |
|---|---|---|
| CAN (ours) | 27.1% | 27.01% |
| Vanilla ANNs | 22.5% | 15.5% |

The above values are the results by using the CIFAR-10 dataset.

## 5.3 TIME TO STABILITY

The time (in terms of epochs or iterations) it takes for the model's performance on a new task to stabilize after learning it. Stability is reached when the model's performance on the new task no longer fluctuates significantly. A shorter time to stability implies that the model adapts quickly to new information, which is crucial for reducing training time and computational resources.

While it converged in 10 epochs for the initial task, the model needed more epochs for the second one. For instance, the second task needed 20 epochs to reach convergence. Despite this increased epoch requirement for later tasks, the model consistently achieved convergence. All of these experiments were conducted with a seed value of 720 to ensure reproducibility.

## 6 FUTURE SCOPE

Future scope for our project can be divided into two different aspects, i.e., implementing a Gating system and allowing the network to grow after it reaches its limitation.

**Gating Mechanism:** This is an important mechanism in order to attain true Continual Learning, as the model needs to automatically identify the kind of task and use the respective sub-network for inference. This can easily be implemented by training an auto-encoder for every new task and measuring the reconstruction error.

**Growing Network:** It is understood that a fixed number of neurons cannot keep on learning forever. Therefore, we need a mechanism using which the network grows once all the existing neurons are exhausted. This allows the network to grow forever and attain actual Continual Learning. One possible approach is to use Reinforcement Learning for the network to learn the rules of growing.

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
