# OpenReview forum: "CAN - CONTINUOUSLY ADAPTING NETWORKS"
_ICLR.cc/2025/Conference — Submitted to ICLR 2025_

### Official Review · Reviewer_Zc1B · 2024-10-27

**Soundness:** 2
**Presentation:** 2
**Contribution:** 2
**Rating:** 3
**Confidence:** 3

**Summary:**

This paper provides a method to prevent catastrophic forgetting, with externally-defined task boundaries, using Hebbian learning to enable/disable neurons that were relevant for a prior task.

**Strengths:**

If there is a good conceptual reasoning behind the proposed use of Hebbian learning (see my comment on the matter under "weaknesses"), the direction explored in the paper may be promising.

**Weaknesses:**

1) While experiments demonstrate that Hebbian learning mechanism prevents catastrophic forgetting, it is not clear why (conceptually) this would be the case. In other words, why does association by activity select the nodes that are relevant for one task while excluding those that belonged to past ones? It may be my missing backgeound in the application of Hebbian learning, but I recommend authors to include a more detailed explanation of this part of the conceotual background for people who are lacking this background.

2) All experiments are basically the same measured with different metrics. One of the metrics would suffice.

3) Task identification and new task recognition is not a trivial task. Without this, it is questionable whether any method can be called to realize continual learning or not. (The authors mention that this is not a challenge they are tackling, but without this ffull continual learning setup the relevance of a destructive adaptation problem is questionable.)

**Questions:**

See point (1) above.

---

> ### Author Response · Authors · 2024-11-25
> **Clarification on the application of Hebbian Learning**
>
> Firstly, thank you for your feedback.
>
> The paper tries to tackle continual learning by selectively training sub-networks (using SGD) for different tasks. These sub-networks are chosen based on the hebbian update matrix computed in every forward pass.
>
> Also, I understand that task recognition might not be a trivial task. We do have certain methods like training a simple encoder-decoder model for every task to measure the reconstruction error and then identify the task as the task with least reconstruction error. We left it to the future scope of the project so that we could research on better ways.

---

### Official Review · Reviewer_CkLD · 2024-11-01

**Soundness:** 1
**Presentation:** 1
**Contribution:** 1
**Rating:** 1
**Confidence:** 4

**Summary:**

The authors proposed a novel algorithm named “CAN” that can dynamically freeze weights to prevent catastrophic forgetting. CAN uses Hebbian rule to measure the importance of neurons with presented tasks. While the dynamic gating of gradient flow onto individual neurons may sound interesting, this study does not provide any convincing evidence that this algorithm can be used in modern deep learning models.

**Strengths:**

The use of Hebbian rule when evaluating the neurons' importance is interesting.

**Weaknesses:**

1. The authors did not provide specific details that are crucial for their study. Specifically, the second equation does not include any indices, which makes its interpretation difficult, and the architecture of a neural network used in this study is not specified.

2. The authors split MNIST into two disjoint tasks. The experimental setup is too simple to evaluate the algorithm properly.

3. Although multiple earlier studies used MNIST to evaluate continual learning algorithms, the authors did not provide any comparison to other studies.

**Questions:**

“CAN” evaluates the importance of individual neurons sequentially, but can the authors evaluate its complexity? Modern deep learning models have a massive number of neurons, and if CAN requires too much computations, it may not be practical.

---

### Official Review · Reviewer_YGtJ · 2024-11-02

**Soundness:** 2
**Presentation:** 1
**Contribution:** 1
**Rating:** 1
**Confidence:** 4

**Summary:**

The paper connects the Hebbian learning with neuron freezing, computing neuron importance through a Hebbian layer and selectively freezing neuron weights based on importance scores. Subsequently, it uses hooks to selectively train and infer, effectively assigning different neurons for each task. The paper combines biological neural mechanisms with continual learning, providing some insights. However, it does not sufficiently summarize previous work, and the novelty of this idea is limited. Additionally, the figures and tables in the paper are quite rudimentary, lacking details on model training and testing. The results are not comprehensively compared with existing methods, which makes them unconvincing.

**Strengths:**

The paper connects the Hebbian learning with neuron isolation, selectively freezing neuron weights based on importance calculation, which provides some valuable insights.

**Weaknesses:**

The writing of this paper is poor, lacking a summary of related work in the field. There are many existing methods that combine Hebbian learning with weight updates, but the paper fails to mention the limitations of current methods or the advantages of the proposed approach. Additionally, the figures and tables are quite rudimentary, making it difficult to understand the authors' intent. The experimental results lack detail, including mathematical descriptions and specific training and testing procedures. More importantly, there is no extensive and comprehensive comparison, and the limited experimental results are insufficient to demonstrate the effectiveness of the model.

**Questions:**

1. There have been many continual learning methods based on Hebb learnin before; what advantages does the proposed method have over these previous works?
2. What are the specific implementations of the model training and testing process, and can they be described in detailed mathematical terms?
3. Are there more comprehensive and detailed comparison results available in the paper?

---

### Official Review · Reviewer_291i · 2024-11-02

**Soundness:** 1
**Presentation:** 2
**Contribution:** 1
**Rating:** 1
**Confidence:** 3

**Summary:**

In this paper, the authors attempt to address the continual learning problem by selectively freezing important neurons when training on a new task. In contrast to previous continual learning methods, important neurons are identified with Hebbian learning using a network of the same architecture. The model is then evaluated on a simple MNIST task and is shown to significantly outperform a vanilla model.

**Strengths:**

The idea of utilizing Hebbian learning to help the model continually learn is interesting, as Hebbian learning is a well-known synaptic plasticity rule in animals. Experiments might provide insights into why these mechanisms can help animals continually learn.

**Weaknesses:**

1. Overall the motivation behind the method is not well explained. The motivation behind Hebbian learning is to take inspiration from animals, but it is not clear to me why we want to use a separate network evolved with Hebbian learning to compute the importance score. It's unclear how this dual-network design is linked to biological continual learning.
2. More importantly, the experiments are way too lacking. The method is only tested on MNIST with 0-4/5-9 as two separate tasks and only compared to the vanilla network. The paper mentioned quite a lot of previous work but they are not compared as baselines. It's understandable that these bio-inspired methods might not surpass SOTA methods in the field but at least the method should be tested on a range of scenarios against some reasonable baseline.

**Questions:**

In addition to the weaknesses mentioned above:
1. How is this linked to previous methods that also utilize important scores and what's the advantage of Hebbian learning here? I also expect more evidence that the computed importance score can actually identify important neurons, for example, how does the method compare to pre-assigning a set of neurons for each task?
2. How is the network tested on different tasks? Is the whole network used in each task or some mask is used even in the forward pass?

---

> ### Author Response · Authors · 2024-11-25
> **Elaborate explanation and correction on the usage of Hebbian Learning**
>
> Thank you for your valuable feedback.
>
> The current explanation in the paper shows how two networks run in parallel where the one using Hebbian learning is used to compute important neurons and that information is used by back-propagation to selectively train the neurons. I realized the obvious error here that since both the networks are not related in any way, the computation of neuron scores doesn't make any sense with respect to the other network.
>
> We have come up with a slight variation of the idea where only a single network is present. The idea here is to use both a global feedback (loss) and local feedback (hebbian updates) similar to how it is in the human brain. After every forward propagation, the hebbian update matrix is computed. The computed matrix is normalized between 0 and 1 so that it could be used as scalar values in further processing. This matrix is then multiplied with the gradients during back-propagation. This scaling selects the important and frequent connections by multiplying the rest by 0 and also scales the important ones w.r.t their importance. The basic idea here would be to give some local feedback to the network and also selectively train for different tasks.
>
> Whole network is not used in the forward pass as of now, each task has its forward mask.
>
> Also, according to the feedback we are testing it on more standard datasets with better experiments.

---

> > ### Comment · Reviewer_291i · 2024-11-26
> >
> > Thanks for the clarification. As of now, the work seems incomplete I would be interested to see improvements to the method and new experiments in the future.

---

### Author Response · Authors · 2024-12-01
**Changes in the Rebuttal Submission**

Following are the changes made by us:

1) Modified the idea of using two separate networks. Instead, now both the algorithms are dealing with a single network and the same parameters. In this version, the gradients of the SGD algorithm are scaled according to the synaptic importance computed using Hebbian learning. This causes the gradients to flow only through strong and frequent connections, that too based on their connection strength.

2) Conducted experiments to test both Task-Incremental and Domain-Incremental related tasks. We conducted experiments using both the MNIST and CIFAR-10 datasets.

3) Added more plots and removed redundant metrics.

Kindly review the new submission and please provide your valuable feedback.

---

### Meta-Review · Area_Chair_Tt9D · 2024-12-17

**Metareview:**

The paper introduces a continual learning framework that leverages Hebbian learning to dynamically freeze network parameters, thereby mitigating catastrophic forgetting. The authors demonstrate that this approach effectively reduces catastrophic forgetting in simple continual learning tasks on the MNIST dataset.

**Strengths** While the use of Hebbian learning for identifying parameter importance has been explored in prior work [A], its biological origins still make it an interesting approach worth further exploration.

**Weaknesses** The major criticisms of this paper include: 1) the absence of rigorous experiments across a range of continual learning benchmarks, the paper has a rudimentary experiments section; 2) insufficient comparison with related works; 3) inadequate motivation and discussion; and 4) a lack of detailed mathematical formulation and experimental setup.

The reviewers unanimously agreed that the paper is incomplete and not ready for publication at ICLR. I completely agree with their assessments and recommend rejection of the paper.

[A] Kolouri, Soheil, Nicholas Ketz, Xinyun Zou, Jeffrey Krichmar, and Praveen Pilly. "Attention-based structural-plasticity." arXiv preprint arXiv:1903.06070 (2019).

**Additional Comments On Reviewer Discussion:**

The authors agreed with the reviewers' assessment of their paper, leading to an uneventful post-rebuttal discussion period.

---

### Decision · Program_Chairs · 2025-01-22

Reject